# BNT162b2 Vaccination after SARS-CoV-2 Infection Changes the Dynamics of Total and Neutralizing Antibodies against SARS-CoV-2: A 6-Month Prospective Cohort Study

**DOI:** 10.3390/vaccines11061127

**Published:** 2023-06-20

**Authors:** Jorge Hernández-Bello, José Julio Sierra-García-de-Quevedo, José Javier Morales-Núñez, Guillermo Santoscoy-Ascencio, Saúl Alberto Díaz-Pérez, Jesús Alberto Gutiérrez-Brito, José Francisco Muñoz-Valle

**Affiliations:** 1Institute of Research in Biomedical Sciences, University Center of Health Sciences (CUCS), University of Guadalajara, Guadalajara 44340, Jalisco, Mexico; jorge.hernandezbello@cucs.udg.mx (J.H.-B.); javier.morales@academicos.udg.mx (J.J.M.-N.); saul.diaz2330@alumnos.udg.mx (S.A.D.-P.); jesus.gutierrez3225@alumnos.udg.mx (J.A.G.-B.); 2Unidad de Patología Clinica (UPC), Guadalajara 44650, Jalisco, Mexico; julio.sierra@upc.com.mx (J.J.S.-G.-d.-Q.); guillermo.coy7@gmail.com (G.S.-A.)

**Keywords:** neutralizing antibodies, SARS-CoV-2, BNT162b2, COVID-19

## Abstract

This study aimed to analyze the dynamics, duration, and production of total and neutralizing antibodies induced by the BNT162b2 vaccine and the possible effect of gender and prior SARS-CoV-2 infection on the generation of these antibodies. Total antibodies were quantified via chemiluminescent microparticle immunoassay (CMIA), and neutralizing antibodies were quantified using the cPass SARS-CoV-2 kit. Individuals with a history of COVID-19 produced twice as many antibodies than vaccinated individuals without prior SARS-CoV-2 infection, with an exponential increase observed in just six days. In those without a COVID-19 history, similar antibody production was reached 45 days after vaccination. Although total antibodies decline considerably in the first two months, the neutralizing antibodies and their inhibitory capacity (>96%) persist up to 6 months after the first dose. There was a tendency for higher total antibodies in women than men, but not at the inhibition capacity level. We suggest that the decline in total antibodies should not be considered as an indicator of loss of protective immunity because most antibodies decay two months after the second dose, but neutralizing antibodies remain constant for at least six months. Therefore, these latter antibodies could be better indicators for estimating the time-dependent vaccine efficacy.

## 1. Introduction

Once the SARS-CoV-2 (COVID-19) pandemic was declared in March 2020, a scientific race began to develop a vaccine that could control the global health crisis [1]. The first vaccine approved for emergency use by the World Health Organization (WHO) was the BNT162b2 (Pfizer-BioNTech) [2], an mRNA vaccine considered effective in preventing infection and hospitalization, even in the face of variants of concern [3,4]. However, the latter is controversial since there are studies that report the contrary [5].

The efficacy of a vaccine can be assessed by both the cellular and humoral immune response [6]. However, for SARS-CoV-2 vaccines, there are still questions about the duration and kinetics of humoral immunity regarding the production of antibodies by B lymphocytes, even for the BNT162b2 one, which was the first approved SARS-CoV-2 vaccine [7].

Antibody generation in response to a viral infection or vaccination is an essential mechanism acquired for viral clearance by adaptive immunity [8,9]; this response can be fast through an extrafollicular stage with the generation of antibodies predominantly of the IgM isotype with little somatic hypermutation but with the ability to neutralize the virus [10,11]. On the other hand, there is a slow response that takes several days in its generation, which is called the follicular stage with the generation of germinal centers, where antibodies with high-affinity IgG isotype are produced by antigen-specific B cells experiencing high somatic hypermutation in the germinal center [12,13].

Structurally, an antibody has two regions, the crystallizable region (Fc) and the fragment antigen-binding fragment (Fab) [14]. The antibodies exert their effector function through the Fc region, using the mechanisms of antibody-dependent cellular phagocytosis (ADCP), antibody-dependent cell cytotoxicity (ADCC), and complement-dependent cytotoxicity (CDC), whose purpose is the elimination of infected cells, and this whole process is knows as antibodies with non-neutralizing action (nNAb) [15,16]. On the other hand, the Fab region exerts the neutralizing function (NAb), providing the reduction in viral infectivity by binding to the virion, thus blocking the process of viral replication [17,18,19].

Therefore, humoral response can be measured in its neutralizing part (neutralizing antibodies) and indirectly in its effector response (non-neutralizing antibodies) employing the total measurement of the antibodies generated against SARS-CoV-2. So far, only a few published articles describe and study both functions simultaneously and longitudinally after SARS-CoV-2 vaccination.

It is essential to clarify that the role of antibody affinity to their antigen is crucial in determining what mechanism they might be performing [20]. However, there are associated factors that could affect the production of antibodies independently of the immune mechanism that activates the vaccination process [21]; these can be intrinsic, such as gender, and extrinsic, such as a pre-vaccination SAR-CoV-2 infection [22]. Regarding gender, studies have shown higher antibody levels against SARS-CoV-2 after vaccination in women [23,24]. The role of the prior infection has been shown to aid in a faster immune response, mainly after the application of the first dose [25]. However, more evidence is needed on the impact of these two factors.

Based on the above, the present study aimed to longitudinally analyze the humoral immune response by the produced antibodies (total or neutralizers) and determine the impact of gender and a prior SARS-CoV-2 infection on this production. In the present study, the levels of anti-SARS-CoV-2 antibodies were analyzed at days 0, 3, 6, 9, 13, 16, 19, 34, 45, 57, 91, and 188 following vaccinations. Therefore, this study is one of the most extensive studies conducted on the dynamics of anti-SARS-CoV-2 antibody production. Due to the large number of determinations, it is possible to show the dynamics of antibody production, the maximal peaks in production, and the beginning of the decay process with precision.

## 2. Materials and Methods

### 2.1. Subjects and Sample Collection

We included 20 volunteer subjects who had been immunized with the BNT162b2 (Pfizer-BioNTech, Leipzig, Germany) vaccine. All subjects were volunteers from the Clinical Pathology Unit (UPC), Guadalajara, Jalisco, Mexico, where they signed an informed consent letter for inclusion in this study. The samples were analyzed at UPC and the University Center of Health Sciences (CUCS) of the University of Guadalajara. The eligibility criteria were persons over 18 years of age, who were without diseases that compromised the immune system, who did not intake immunosuppressive drugs, and non-pregnant women. Subjects were divided into two groups: (a) immunized subjects without prior COVID-19 infection (*n* = 9) and (b) immunized subjects with prior COVID-19 infection (*n* = 11). All volunteers completed the study; there were no dropouts.

The study was conducted from March to September 2021, where peripheral blood samples were taken via a venous puncture in vacutainer tubes without an anticoagulant to obtain serum on days 0, 3, 6, 9, 13, 16, 19, 34, 45, 57, 91 and 188 after receiving the first dose of BNT162b2. The second BNT162b2 dose was applied 35 days after the first dose. Surveys were conducted with the participants to obtain clinical and demographic data at baseline and throughout the follow-up. Participants during the study follow-up did not receive any additional doses (booster) to complete their vaccination schedule of BNT162b2. This study was conducted in accordance with the Declaration of Helsinki and the Mexican Standard NOM-012-SSA3-2012 that establishes the criteria for the execution of research projects on health involving human beings.

The group previously diagnosed with COVID-19 included participants with confirmatory RT-PCR (real-time reverse transcription–polymerase chain reaction) test results for SARS-CoV-2 infection 7 to 9 months before the first dose of vaccine. For both groups, IgG/IgM anti-SARS-CoV-2 antibodies were analyzed on day 0 to corroborate a prior infection.

### 2.2. Detection of IgM/IgG against SARS-CoV-2

The presence or absence of IgM/IgG antibodies against SARS-CoV-2 was determined using the Certum IgM/IgG Rapid Test cassette kit (Certum Diagnostics, Nuevo León, Mexico). This kit reacts to the presence of nucleocapsid (N) and spike (S) proteins. It is a qualitative test and detects the fractions separately. The protocol was performed according to the manufacturer’s instructions. This test is based on the principle of lateral flow chromatographic immunoassay.

### 2.3. Quantification of Total Antibodies against SARS-CoV-2

Total antibodies against SARS-CoV-2 (IgG) were measured using the Chemiluminescent Microparticle Immunoassay (CMIA) technique in the ABBOTT brand Alinity equipment (SARS-CoV-2 IgG II Quant). The quantitative determination of IgG antibodies is directed against the S1 region of the “spike” protein, specifically the receptor-binding domain (RBD) site. The cut-off point of the test is 50 AU/mL. Values below this level are considered negative; values equal to or above this point are considered positive. Quantitatively the lower limit value of measurement is 0.0 AU/mL and the upper limit of determination in direct measurement is 40,000 AU/mL which can be extended to 80,000 AU/mL by a 1:2 dilution. Based on the manufacturer’s recommendations, quantitative measurement was used for this research.

### 2.4. Quantification of Neutralizing Antibodies against SARS-CoV-2

Quantification of neutralizing antibodies was performed on all serial samples (day 0 to 188 after the application of the BNT162b2 vaccine) using cPass™ SARS-CoV-2 Neutralization Antibody Detection Kit (GenScript, Piscataway, NJ, USA), based on the blocking enzyme-linked immunosorbent assay (ELISA). The neutralization test was performed according to the manufacturer’s instructions. The result is interpreted as positive if the value is equal to or greater than 30% of the inhibition of the signal. The inhibition rate was calculated using the following formula: $$\%\;neutralization=\left(1-\frac{OD\;value\;of\;Sample}{OD\;value\;of\;Negative\;Control}\right)\times 100\%$$

## 3. Results

### 3.1. Clinical Characteristics of Study Groups

The clinical and demographic characteristics of both study groups are shown in Table 1. No significant differences were observed between groups regarding age, gender, comorbidities, or treatments (*p* > 0.05).

### 3.2. Neutralizing Antibodies Evolution

All patients were followed-up after they were vaccinated with the first BNT162b2 dose. Blood samples were drawn on days 0, 3, 6, 9, 13, 16, 19, 34, 45, 57, 91, and 188 to determine the presence of neutralizing antibodies and their neutralization capacity. Figure 1 shows the medians of the neutralizing capacity percentage of anti-SARS-CoV-2 antibodies after vaccination with the BNT162b2 vaccine in all the individuals included in this study (*n* = 20). A significant difference in the neutralization percentage from day 3 to 6 (*p* = 0.0004), day 6 to 9 (*p* = 0.0056), and day 34 to 45 (*p* = 0.0329) was observed. However, it is remarked that globally, the neutralization percentage increased from day 3 to 6 after vaccination, having only slight variations in the percentage of neutralization from day 6.

In addition, study groups were divided by gender and COVID-19 history status to acknowledge the effect of these two characteristics on the percentage of neutralizing antibodies. In the case of gender (Figure 2), the overall evolution of neutralizing antibodies percentage showed no difference between male and female patients, except for day 13, as men had antibodies with a slightly lower neutralizing capacity (percentages) than women (95.22 vs. 97.20, *p* = 0.0415).

Regarding COVID-19 history (Figure 3), differences in the percentage of neutralizing antibodies between the two groups were significant on days 9 (55.51 vs. 96.64, *p* = 0.0432), 13 (97.20 vs. 76.20, *p* = 0.0077), 16 (80.73 vs. 96.57, *p* = 0.0101) and 19 (84.35 vs. 96.95, *p* = 0.0135). Notably, the neutralizing capacity of the antibodies of people with a previous SARS-CoV-2 infection reached percentages of >90% on day nine after the first BNT162b2 dose. In contrast, in individuals without prior COVID-19, the infection reached a similar percentage on day 34, having a maximum neutralizing peak at 11 days after the second vaccine dose.

### 3.3. Total Antibodies Evolution

SARS-CoV-2 total antibodies were also measured in the same patients for the same period (Figure 4). An increase in the total anti-SARS-CoV-2 antibodies concentration was observed after a BNT162b2 dose. Significant changes are shown between days 3 and 6 (*p* = 0.0024), days 6 and 9 (*p* < 0.0001), and days 34 and 45 (*p* = 0.0005). It is noteworthy that on day nine, the highest level of total antibodies is reached.

A decrease after the administration of any dose of the BNT162b2 vaccine is observed after 9–12 days, followed by a significant decrease for days 9 to 13 (*p* = 0.0219), 13 to 16 (*p* = 0.0309), 19 to 34 (*p* = 0.0001), 45 to 57 (*p* = 0.0019), 57 to 91 (*p* = 0.0039) and 91 to 188 (*p* = 0.0312).

Additionally, the total anti-SARS-CoV-2 antibodies evolution was analyzed by gender (Figure 5). In general, total antibody levels were higher in women than in men, however, a significant difference only on day 45 was observed (23,146 vs. 37,657, *p* = 0.0480), though total anti-SARS-CoV-2 antibodies were higher in females (*p* < 0.05).

Segregating the patients by prior COVID-19 status (Figure 6) showed differences between total antibodies on days 6 (0.6 vs. 17,117, *p* = 0.0477), 13 (1137 vs. 42,221, *p* = 0.0145), 16 (1592 vs. 33,513, *p* = 0.0247) and 19 (2556 vs. 32,220, *p* = 0.0429); these last three differences agree with those observed in the neutralizing capacity comparison. It is important to note that the number of antibodies produced in patients with a prior history of COVID-19 is double than that produced in vaccinated patients without a history of prior SARS-CoV-2 infection. In addition, the production of total antibodies against SARS-CoV-2 is observed from day 6 in patients with a history of COVID-19, reaching the maximum peak at day 16. In contrast, in individuals without previous COVID-19, a significant increase in antibody production is reached on day 45, which coincides with the maximum peak.

### 3.4. Correlation between Total Anti-SARS-CoV-2 Antibodies and Neutralizing Capacity

Table 2 shows a high and significant correlation between total anti-SARS-CoV-2 antibodies and neutralizing capacity from day 0 until day 34; this correlation is lost from days 45 to 188.

## 4. Discussion

The BNT162b2 vaccine produces antibodies specifically against protein S [26], which is a large protein (450 kDa as a trimer) with conserved and no conserved sequences that can generate antibodies with neutralizing and non-neutralizing action [27,28].

We observed that total antibodies against SARS-CoV-2 (Figure 4) begin to increase on day three to six, reaching their peak on day nine post-vaccination of the first BNT162b2 dose. Thereafter, these antibodies began to drop rapidly. Ten days after the second dose (day 45), antibodies increased again, but from this point, they gradually declined, which was significantly low at day 188 from the first-dose application. Additionally, antibodies with neutralizing capacity (Figure 1) also presented a significant increase from day 6. Still, contrary to that observed in total antibodies, the neutralizing capacity remained high until day 188. The concordance in the first month post-vaccination between total and neutralizing antibodies is evident in Table 2. However, after day 45, this concordance disappears. From this result, it can be inferred that it is possible that effector antibodies (which have a more significant role in resolving an active infection) are the first to decay, while antibodies with neutralizing capacity persist.

Đaković Rode et al. reported an exponential decrease in total anti-SAR-CoV-2 antibodies three and six months after completing the vaccine scheme (two doses) [29]. Bayart et al. reported a gradual decrease for both types of antibodies at six months of vaccination [30], which matches our total antibodies but contrasts with that observed for neutralizing antibodies, as we report them stable for the same time.

Regarding neutralizing antibody dynamics in response to the BNT162b2 vaccine, Terpos et al. reported a decrease after six months [31], which is consistent with what was reported by Bayart but is in contrast with our results. On the contrary, Muena et al. reported the persistence of neutralizing antibodies against SARS-CoV-2 until 13 months after vaccination with the BNT162b2 or CoronaVac vaccines [32], which is consistent with our findings. Other studies also report the persistence of neutralizing antibodies for several months, but with a decreasing trend in their inhibitory capacity [33,34,35,36].

Our results and previous reports agree that total antibodies decrease quickly over time. The fraction that mainly decreases may correspond to non-neutralizing antibodies, which carry out effector functions, such as ADCP, ADCC, and CDC, all of which are essential mechanisms for eliminating infected cells [28,37]. However, the persistence of these antibodies before primary infection or reinfection, could develop an antibody-dependent enhancement (ADE) in the individual, a phenomenon that increases the virulence of antibody-mediated pathogens [38]. Therefore, the rapid decrease in these antibodies could be a protective mechanism against ADE.

Regarding gender, we observed that both women and men show a similar dynamic for total antibody synthesis and decay. However, 45 days (10 days after the second dose of BNT162b2) post-vaccination, women present significantly higher levels of antibodies than men. Figure 5 shows a trend of higher total antibodies in women than in men. However, for neutralizing antibodies, we only observed a significant difference on day 13, observing more neutralizing antibodies in women than in men, but in general, there were no significant differences in the inhibitory capacity by gender.

On the other hand, Markmann et al. reported that men produce more total and neutralizing antibodies than women [39], which contrasts with our results. Nevertheless, in a longitudinal study (3 months) performed by Heriyanto et al., they observed that women produced more neutralizing antibodies than men after vaccination against SARS-CoV-2 [40]. The differences in antibody responses of males and females have been well documented. Both genetic and hormonal factors likely influence these sex differences in humoral immunity. Research suggests that estrogen enhances specific immune responses mediated by B cells [41]. Therefore, on average, studies report a preference for the female gender to produce greater antibody responses and higher B cell activity compared to men [23,42,43]. However, other authors have determined that gender does not affect the production of antibodies induced by natural infection or vaccination [44,45]. Numerous factors can influence immune function, including age, underlying health conditions, and lifestyle; therefore, more studies are needed in this regard, especially in larger cohorts.

One of our most interesting findings was the significant differences in the speed of antibody synthesis (days) after vaccination between individuals with and without a SARS-CoV-2 history. Six days post-vaccination, patients with a history of COVID-19 reach a high antibodies level, contrasting significantly with that observed in those without a COVID-19 history, which reach a significant antibodies synthesis 45 days post-vaccination (10 days after the second vaccine dose). The same dynamics were observed regarding neutralizing antibodies and their inhibitory capacity; individuals with a COVID-19 history reached inhibitory capacity percentages >90% at day 9 after the first BNT162b2 dose, while in individuals without prior COVID-19, it was 34 days post-vaccination.

Gobbi et al. reported that people with prior infection generated more total antibodies than those without prior infection; they also observed that after seven days of the second BNT162b2 dose, both groups increased these antibody levels and matched their production. For neutralizing antibodies, even after the second dose, the group with prior infection continued to produce more neutralizing antibodies [46].

Carbonare et al. conducted the same analysis with both groups in another population. They also reported more neutralizing antibody production in people with prior COVID-19, even after the second dose [47]. Decru et al. studied the dynamics of neutralizing antibodies induced by the BNT162b2 vaccine for ten months; they concluded that people with a prior infection maintain their antibodies with neutralizing capacity, unlike those vaccinated groups without prior infection or people not vaccinated with a natural infection [34].

Finally, we clarify that the main limitation of the present study is that we did not perform any techniques to evaluate the non-neutralizing function of antibodies generated against SARS-CoV-2 in response to BNT162b2. We only extrapolated the information based on the behavior of total antibodies; however, this weakness is a perspective for future studies. In future studies, assessing the neutralization dynamics of specific interest variants due to the emergence and establishment of new variants is vital. Another limitation could be the sample size (*n* = 20); however, it is essential to highlight that due to the high number of samples required for each patient, it is complex to follow up with a large number of individuals.

## 5. Conclusions

The BNT162b2 vaccine induces high levels of antibodies (neutralizing and non-neutralizing antibodies) in individuals with or without previous COVID-19. However, individuals with a history of COVID-19 produce twice as many antibodies, with an exponential increase in just 6 days, while those without COVID-19 history reached a similar antibody production 45 days after vaccination. Although these antibodies decline considerably in the first two months, the neutralizing antibodies and their inhibitory capacity (>90%) persist up to 6 months after the first dose. Sex-based differences in antibody production were not significant, but a tendency for a better antibody response in women was observed.

## Figures and Tables

**Figure 1 vaccines-11-01127-f001:**
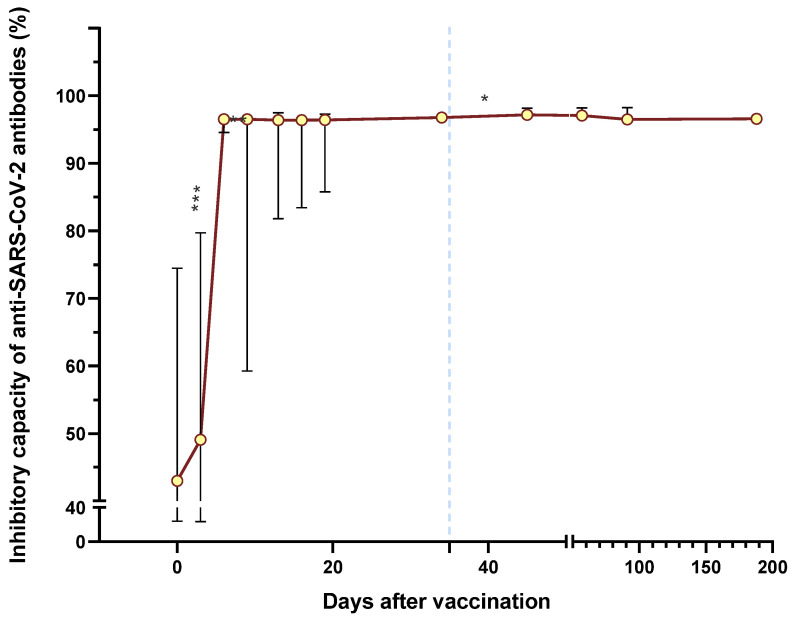
Medians of the neutralizing capacity percentage of anti-SARS-CoV-2 antibodies after vaccination with the BNT162b2 vaccine in all the individuals included in this study (*n* = 20). The second dose application is represented with a segmented blue line at day 35. The difference in medians was determined using Wilcoxon’s signed rank test. * *p* < 0.05; *** *p* < 0.001.

**Figure 2 vaccines-11-01127-f002:**
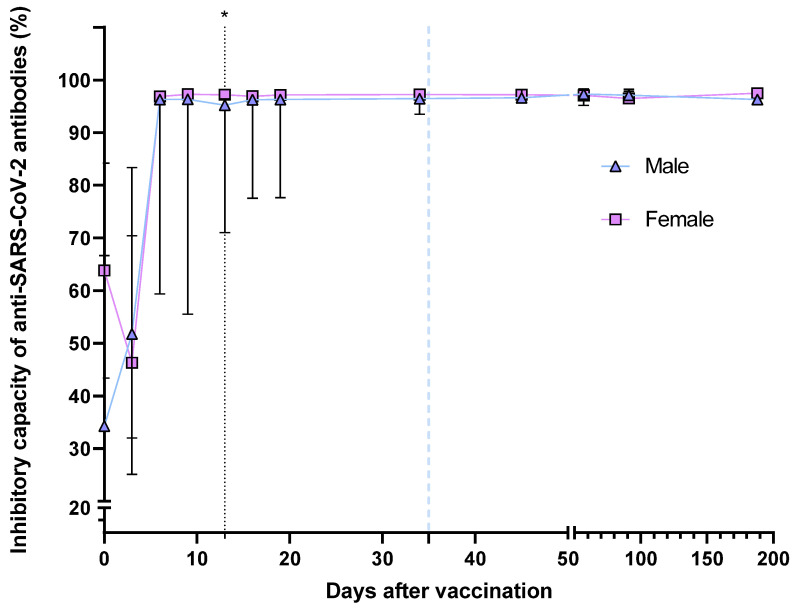
Medians of the neutralizing antibodies percentage after the BNT162b vaccination, divided by gender. The second dose application is represented with a segmented blue line at day 35. The difference in medians was determined via Mann–Whitney’s U test. * *p* < 0.05.

**Figure 3 vaccines-11-01127-f003:**
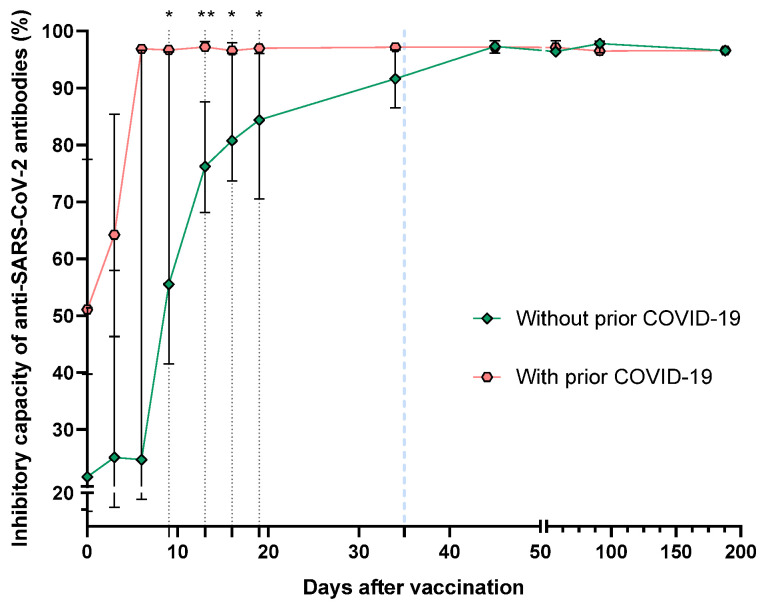
Medians of the neutralizing antibodies percentage after vaccination with the BNT162b2 vaccine, divided by prior COVID-19 status. The second dose application is represented with a segmented blue line at day 35. The difference in medians was determined using Mann–Whitney’s U test. * *p* < 0.05; ** *p* < 0.01.

**Figure 4 vaccines-11-01127-f004:**
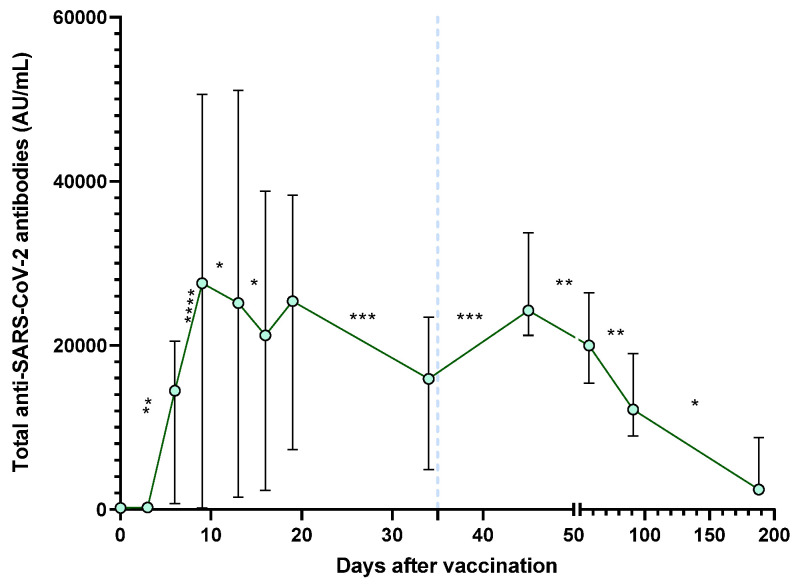
Medians of total antibodies after vaccination with BNT162b2 vaccine. The second dose application is represented with a segmented blue line at day 35. The difference in medians was determined using Wilcoxon’s signed-rank test. * *p* < 0.05; ** *p* < 0.01, *** *p* < 0.001; **** *p* < 0.0001.

**Figure 5 vaccines-11-01127-f005:**
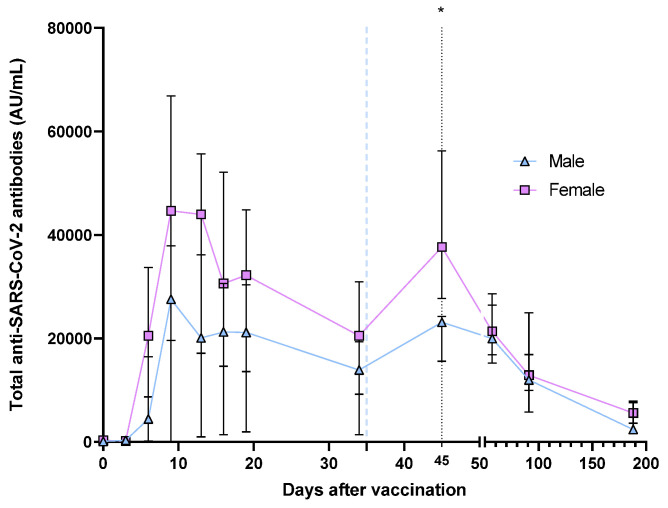
Medians of the total antibodies after vaccination with the BNT162b2 vaccine, divided by gender. This analysis was performed on the total individuals (*n* = 20). The second dose application is represented with a segmented blue line at day 35. The difference in medians was determined via Mann–Whitney’s U test. * *p* < 0.05.

**Figure 6 vaccines-11-01127-f006:**
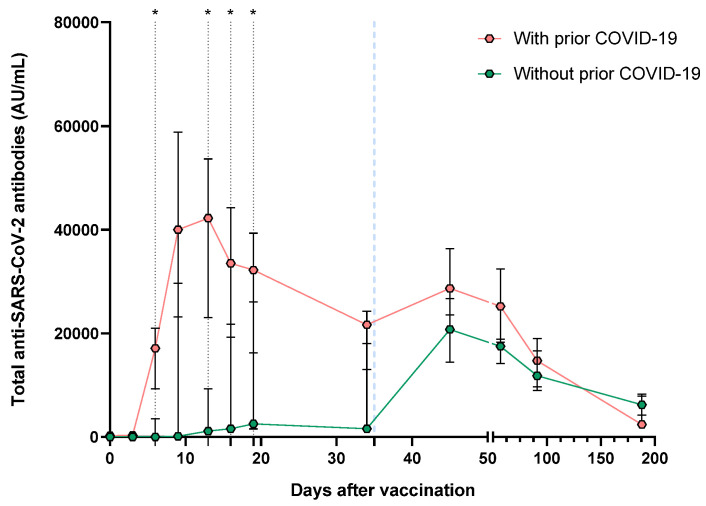
Medians of total antibodies evolution after vaccination with BNT162b2 vaccine, divided by prior COVID-19 status. The second dose application is represented with a segmented blue line at day 35. The difference in medians was determined using Mann–Whitney’s U test. * *p* < 0.05.

**Table 1 vaccines-11-01127-t001:** Sociodemographic information from the study groups.

	Immunized with the BNT162b2 Vaccine		*p*-Value
	Without Prior COVID-19 (*n* = 9)	With Prior COVID-19 (*n* = 11)	
Age (years), ^mean ± SD^	42.67 ± 15.64	36.91 ± 12.62	0.373
Gender, *^n^* ^(%)^			
Female	2 (22.22)	6 (54.55)	0.374
Male	7 (77.78)	5 (45.45)	
Comorbidities, ^*n* (%)^			
At least one	6 (6.66)	5 (45.45)	1.000
Overweight/obesity	3 (3.33)	4 (36.36)	1.000
Allergic diseases	4 (44.44)	2 (18.18)	0.335
SAH	2 (22.22)	2 (18.18)	1.000
Thyroid disease	1 (11.11)	1 (9.09)	1.000
Diabetes	0 (0)	1 (9.09)	1.000
Treatment, *^n^* ^(%)^			
At least one	4 (44.44)	6 (54.54)	1.000
Vitamin D	3 (33.33)	2 (18.18)	0.616
Analgesics	2 (22.22)	3 (27.27)	1.000
NSAIDs	0 (0)	1 (9.09)	1.000
Antihypertensive	2 (22.22)	1 (9.09)	0.565
Hypolipidemic agents	1 (11.11)	2 (18.18)	1.000
Beta-blockers	2 (22.22)	0 (0)	0.189
Synthetic hormones	1 (11.11)	1 (9.09)	1.000
Antiacids	1 (11.11)	0 (0)	0.450
Supplements	1 (11.11)	0 (0)	0.450
Antihistamines	0 (0)	1 (9.09)	1.000
Hypoglycemic agents	0 (0)	1 (9.09)	1.000
Contraceptives	0 (0)	1 (9.09)	1.000

Abbreviations: SAH—Systemic Arterial Hypertension; NSAIDs—Non-Steroidal Anti-Inflammatory Drugs. Differences between means were calculated using the Student’s T test, and frequencies were compared via Fisher’s exact test.

**Table 2 vaccines-11-01127-t002:** Correlation between neutralizing antibodies and total antibodies.

Days after the First Dose of the BNT162b2 Vaccine	Spearman’s rho Value	*p*-Value
0	0.9690	7.4337 × 10^−6^
3	0.9617	2.9523 × 10^−9^
6	0.8044	9.9007 × 10^−5^
9	0.7762	0.0002
13	0.8827	1.2358 × 10^−6^
16	0.8837	5.2748 × 10^−7^
19	0.8543	3.2399 × 10^−6^
34	0.8114	0.0002
45	−0.0665	0.83719
57	0.3713	0.2347
91	0.2895	0.2952
188	0.2142	0.6615

Correlations were calculated using Spearman’s correlation test.

## Data Availability

The datasets used and/or analyzed during the current study are available from the corresponding author on reasonable request. Informed consent was obtained from all subjects involved in the study. Written informed consent has been obtained from the patient(s) to publish this paper.

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
