# Peer review of "BNT162b2 Vaccination after SARS-CoV-2 Infection Changes the Dynamics of Total and Neutralizing Antibodies against SARS-CoV-2: A 6-Month Prospective Cohort Study"

_vaccines, 2023, doi:10.3390/vaccines11061127_

Round 1

Reviewer 1 Report

This manuscript investigated the antibody dynamics after BNT162b2 injection. Different factors, such as gender and w/ or w/o prior infection were considered. The manuscript is clear, but two main issues need to be fixed before considering publication.

(1)   In all figures, the antibody levels were presented as medians. However, it is more common and reasonable to present the individual data points with error bars.

(2)   For the significant test, were they one-tailed or two tailed?

Author Response

We thank you in advance for the contributions of each reviewer and the editor to improve our article. We describe the changes made below.

Reviewer 2 Report

The manuscript by Hernandez-Bello et al., characterizes the antibody responses generated by immunization with the BNT162b2-based CoV-2 vaccine in two cohorts of adults, (1) a group of 9 individuals with prior Covid infections and (2) a group of 11 without prior Covid infection.  Immune responses were characterized for both total antibody (Ab) and neutralizing Ab responses.  As noted by the authors (L69-71), "the present study aimed to longitudinally analyze the humoral immune response by the produced antibodies (total or neutralizers) and determine the impact of gender and a prior infection by SARS-CoV-2 on the production of these". 

A reason for this study offered by the authors (L57-59) is that, "So far, few published articles describe and study both functions simultaneously and longitudinally after SARS-CoV-2 vaccination".  While this manuscript is fundamentally sound, it is not clear what new would be provided to investigators/clinicians by its publication.  One immediate response to the reasoning in submitting the work is this does not seem consistent with what has already been published.  To wit, there are ~45 articles on PubMed that should have been available to the authors during the preparation of this manuscript (pre-2023) (PubMed search terms: cov 2 AND neutral* AND antibod* AND durab* AND BNT162b2).  Even if the intent of the authors is provide additional weight of evidence to one side of an argument or the other, it is the onus of the authors to concisely and effectively summarize what the conundrum is (e.g., total Ab or neutralizing Ab).  This manuscript does not accomplish this.

It seems to be the consensus of the field that neutralizing Ab are a more effective metric of immune-mediated protection than total Ab titers.  It also seems to be the consensus of the filed that a salient question is whether vaccination with the prototypical CoV-2 sequence confers protection against subsequent genetic variants.  This manuscript does not evaluate protection against subsequent genetic variants. 

The authors raise the valid point (Discussion L239-246) that, "The fraction [total antibodies] that mainly decreases may correspond to non-neutralizing antibodies, which carry out effector functions such as ADCP, ADCC, and CDC, all essential mechanisms for eliminating infected cells [16], [25]; however, the persistence of these antibodies before primary infection or reinfection, could develop in the individual an antibody-dependent enhancement (ADE), a phenomenon that increases the virulence of antibody-mediated pathogens [26]. Therefore, the rapid decrease in these antibodies could be a protective mechanism against ADE".  This is absolutely correct, and the role(s) of non-neutralizing Ab is significantly understudied for CoV-2 and other pathogens. Yet, this aspect of the immune responses was not evaluated by the authors, and this could have had an impactful contribution to the field.  As such, it seems like a missed opportunity.  It may be that the rapid decline of these putative erector functions could be a mechanistic basis for reinfection, which  remain completely unresolved.

As stated, there are no fundamental concerns about the methodologies and presentation of the data.  However, the conclusions are further undercut by what seem like misstatements about basic B-cell responses to prior infections.  For instance (Discussion L262-264), the authors write that, "One of our most interesting findings was the significant differences in the rate of antibody synthesis between individuals with and without SARS-CoV-2 infection prior to vaccination".  The presence of high avidity, memory B cells arising from prior exposure to antigen, either through prior infection and/or prior immunization, ensures that the response to subsequent antigen encounter(s) will be exceedingly rapid.  Thus, it is not clear why the authors vies this phenomenon as "most surprising".  The surprise would have been if rapid memory responses were not observed.

Additional Comments

1) For the cohort with prior CoV-2 infection, it would be useful to listnthe interval of time from prior infection to the time of vaccination.

2) Results L163: The authors note that oscine responses become "optimized" 11 days after vaccination.  What is meant by "optimized"?

3)  Results L183: The authors note that vaccine responses tended higher in women than in men.  It would be useful to offer some speculations why this might be.

Author Response

(The authors gave the same response as above.)

Reviewer 3 Report

General overview

The efficacy of all vaccines against SARS-CoV-2, but especially of those based on the new mRNA platform such as BNT162b2, has regained a worldwide significance from the moment they were launched. Humoral immune response is essential for viral clearance, and can be evaluated by determination of total, anti-RBD and neutralizing antibodies. Longitudinal dynamics of each component of the humoral response is important to evaluate efficacy of vaccination in both naive and previously infected individuals of both genders, age categories, etc. However, in order to achieve valid and comparable results, methodology must be flawless.

 Major comments

My primary concerns are related to the methodology – study sample in particular. Sample is small – consisting of only 20 participants – and furthermore, details on the recruitment of participants (were they employed, students, family members, associates, community members, patients, or else?), drop-outs, lost to follow-up, etc. are simply lacking. Accordingly, subgroups (according to gender and previous covid19 status) are of very small size, unabling any stronger statistical analysis and hence, valid conclusions.

Also, why were so many time points for each subject? 12 time points for blood sampling? What was the drop-out rate?

L97: Is this test constructed for the detection of total anti-N and anti-S antibodies, or is the each fraction (IgM, IgG) detected separately? Also, is this test merely qualitative (present/absent), semi-quantitative or quantitative? These points are important to complete this section in Methodology.

L103: It is unclear if all obtained serum samples were also analyzed using this test or only the serums tested positive with the Rapid test?

L104: You cannot claim that the test is meant for the quantification of total antibodies since it is constructed only for the quantification of anti-S1 IgG antibodies.

L283-289: Strengths and limitations section should be prolonged significantly.

Minor comments

L40: Please change 'against' with 'aquired for'.

L48: Please change to 'antigen-binding fragment'.

L51: Please rephrase an entire sentence to provide clarity.

L53: Please delete the 'defining it as' and replace it with 'providing'.

L56: Please replace 'as' with 'and'.

L65: Please replace 'antibodies' with 'antibody levels'.

L67: Please replace 'in' with 'after'.

L71: Please change to 'a prior SARS-CoV-2 infection'.

L78: Please change 'people' to 'persons'.

L81: Very small groups for this type of analysis.

L85: Why were so many time points for each subject (12 time points for blood sampling)? What was the drop-out rate?

L91: Please change 'proyects' to 'projects'.

L93-95: The entire sentence needs to be rewritten for the sake of clarity, e.g. The group previosly diagnosed with COVID-19 encompassed the participants with RT-PCR confirmatory test result for SARS-CoV-2 infection 7 to 9 months before the first dose of vaccine.'

L135: Why the medians? Please explain the choice of statistical methods used in the study.

L158: Please change 'significative' to 'significant'.

L160-L163: The entire sentence should be re-written to gain clarity. It is unclear how come the >90% of neutralizing ab was achieved just before the 2nd dose of vaccine (34 days after the 1st dose)?

L207-208: Potential explanation of this finding?

L217: Please change 'begin' to 'began'.

L249: Please change 'greater number' with 'higher levels' or similar.

L283-289: Strengths and limitations section should be prolonged.

L291: 'a high number' should be changed to 'high levels' or similar.

Minor editing of English language required.

Author Response

(The authors gave the same response as above.)

Round 2

Reviewer 2 Report

The authors have effectively responded to the prior critiques.   While the authors acknowledge that one limitation of the study is the small number of study subjects, the results warrant publication to add the collective body of knowledge about vaccine-induced antibody responses in those with or without prior covid infection.

Reviewer 3 Report

The authors have addressed all my previous comments.

Minor English language editing required.